# Genome-Wide Identification and Analysis of the GRAS Transcription Factor Gene Family in *Theobroma cacao*

**DOI:** 10.3390/genes14010057

**Published:** 2022-12-24

**Authors:** Sijia Hou, Qianqian Zhang, Jing Chen, Jianqiao Meng, Cong Wang, Junhong Du, Yunqian Guo

**Affiliations:** 1Center for Computational Biology, National Engineering Laboratory for Tree Breeding, College of Biological Science and Technology, Beijing Forestry University, Beijing 100083, China; 2Chinese Institute for Brain Research, Beijing 102206, China; 3College of Biological Science, China Agricultural University, Beijing 100193, China

**Keywords:** *Theobroma cacao*, GRAS transcription factor, plant growth and development, genome-wide analysis, syntenic analysis, gene duplication

## Abstract

*GRAS* genes exist widely and play vital roles in various physiological processes in plants. In this study, to identify *Theobroma cacao* (*T. cacao*) *GRAS* genes involved in environmental stress and phytohormones, we conducted a genome-wide analysis of the GRAS gene family in *T. cacao*. A total of 46 *GRAS* genes of *T. cacao* were identified. Chromosomal distribution analysis showed that all the *TcGRAS* genes were evenly distributed on ten chromosomes. Phylogenetic relationships revealed that GRAS proteins could be divided into twelve subfamilies (HAM: 6, LISCL: 10, LAS: 1, SCL4/7: 1, SCR: 4, DLT: 1, SCL3: 3, DELLA: 4, SHR: 5, PAT1: 6, UN1: 1, UN2: 4). Of the *T. cacao GRAS* genes, all contained the GRAS domain or GRAS superfamily domain. Subcellular localization analysis predicted that TcGRAS proteins were located in the nucleus, chloroplast, and endomembrane system. Gene duplication analysis showed that there were two pairs of tandem repeats and six pairs of fragment duplications, which may account for the rapid expansion in *T. cacao*. In addition, we also predicted the physicochemical properties and cis-acting elements. The analysis of GO annotation predicted that the *TcGRAS* genes were involved in many biological processes. This study highlights the evolution, diversity, and characterization of the *GRAS* genes in *T. cacao* and provides the first comprehensive analysis of this gene family in the cacao genome.

## 1. Introduction

Abiotic stresses including high temperature, drought, cold, and salt have important effects on plant development and growth. Some transcription factors regulate the transcript levels of their target genes under stress by binding to specific DNA sequences in their target promoters [1,2,3]. Therefore, the regulatory networks of various biological processes can be understood by studying plant transcription factors.

The GRAS transcription factor gene family is named after three originally identified members: gibberellic acid insensitive (GAI), GA1 repressor of GA1 (RGA), and scarecrow (SCR) [4,5,6]. Most GRAS proteins contain only one domain, and the C-terminal region of this domain is highly conserved, which is often referred to as the GRAS domain. It contains five units: SAW, leucine heptapeptide repeat I (LHRI), VHIID, PFYRE, and leucine heptapeptide repeat II (LHRII) [7]. However, some GRAS proteins still contain one domain and another functional domain or two domains [7]. The high variability of the N-terminal region of GRAS proteins determines the regulatory proteins’ specificity [8]. The GRAS transcription factor family has been found in many plants, but the family classification varies slightly in different species [9]. For example, in *soybean* (*Glycine max*), 117 *GRAS* genes are divided into 11 subfamilies: AtSCL4/7, Os19, Os4, HAM, DELLA, DLT, AtPAT1, LISCL, AtSCR, AtSCL3, and AtSHR [10]. In *tomato* (*Solanum lycopersicum*), 53 *GRAS* genes have been identified and classified into 13 subfamilies: HAM, LAS, SCL4/7, SCR, SCL9, SCL28, DELLA, SHR, PAT1, Os4, Os19, GRAS37, and Pt20 [11]. In *Brachypodium distachyon* (*Brachypodium distachyon (Linnaeus) P. Beauvois*), 63 GRAS genes are classified into 10 subfamilies: HAM, PAT1, SHR, DELLA, SCL3, SCL4/7, LAS, SCR, DLT, and LISCL [12]. In *potato* (*Solanum tuberosum* L.), 52 GRAS genes have been identified and classified into 8 subfamilies: DELLA, LAS, HAM, PATI, SCR, LISCL, SHR, and SCL3 [13]. However, in *Populus trichocarpa* (*P. trichocarpa*), 93 GRAS genes are divided into 13 subfamilies: Os19, HAM, Os4, Pt20, DLT, AtSCl3, AtSHR, AtPAT1, AtSCR, AtSCL4/7, AtLAS, DELLA, and LISCL [9]. These studies suggest that members of the GRAS gene family differ in different plants. Their classification reflects their evolutionary history.

The diversity of the *GRAS* gene family is consistent with its high diversity of functions [14]. To date, functional identification of GRAS transcriptional regulators has occurred mainly in *Arabidopsis thaliana* and *rice*. This has shown that they have important roles in the regulation of plant development and growth, including multiple growth regulatory signals and environmental signals such as abiotic/biotic stresses, light, and phytohormones [15]. Sun et al. [16] reviewed the major biological functions of the ten subfamilies of GRAS transcriptional regulators, and the results are reported in Table 1.

Although the GRAS gene family has been studied for many years, the mechanisms and evolutionary dynamics of this gene family in woody plants are still not fully understood. Differences in loss and retention of duplicated gene family members between woody and herbaceous species may help in identifying genes with specialized roles in the adaptive evolution of different lineages. The cacao is called “soft gold” because of its high value. The flowers of cacao trees have ornamental value, and cacao is the main ingredient in chocolate and cacao powder [27]. In addition, cacao beans have important uses in the pharmaceutical and cosmetic industries. Cacao is receiving increasing attention for its potential health benefits because it is rich in polyphenols, particularly flavonoids [28]. However, cacao production is hampered for a number of reasons. Therefore, it is of great value to study the cacao tree. The phylogenetic relationship, conserved domain, and collinearity analysis of this family can provide new ideas for further functional analysis. In 2010, Corti et al. [29] carried out the sequencing and assembly study of the *T. cacao* genome, and since then, researchers have successively identified and analyzed its important gene families, such as the NAC gene family [30], WRKY gene family [31], and the GPX transcription factor family [32]. In 2013, researchers completed an analysis of the metabolome and transcriptome of the cacao tree [33]. Our knowledge about the expansion and diversification of this gene family in plants is presently limited to the herbaceous species Arabidopsis. To date, the GRAS gene family has not been identified and classified in *T. cacao*.

In this study, we identified 46 GRAS gene family members and conducted a comprehensive genome-wide analysis of the *GRAS* gene family of the cacao tree, including gene structure, domain analysis, intron/exon, chromosome location, subcellular localization, and cis-acting elements of *GRAS* genes. In addition, we analyzed the phylogenetic relationship of GRAS proteins between *T. cacao* and *A. thaliana*. Furthermore, we performed the gene duplication pattern of *T. cacao* GRAS proteins, and we analyzed a syntenic analysis of GRAS proteins among *T. cacao*, *A. thaliana*, *P. trichocarpa*, and *Sesamum indicum* (*S. indicum*). The results of this study lay the foundation for further studies of the biological function of genes in *T. cacao* and provide a reference for subsequent molecular mechanisms.

## 2. Materials and Methods

### 2.1. Identification of GRAS Gene Family in T. cacao

The genome file, protein file, coding sequences (CDS), and annotation files of *T. cacao* were obtained from Ensembl Plants (http://plants.ensembl.org/index.html, accessed on 1 June 2022). The Hidden Markov Model (HMM) profile of the GRAS protein domain was downloaded from the Pfam protein family database (release 35.0; http://pfam.xfam.org/, accessed on 1 June 2022) under the accession number ‘PF03514’ [34,35].

The HMM model of HMMER (version 3.1b2) was used to screen GRAS protein candidate members of *T. cacao* twice to determine the final target members. Firstly, the downloaded HMM profile was employed using the HMMER v3.3.2 program to search for proteins containing target GRAS domains as the initial filtering results, and ClustalW (version 2.1) was used to perform multiple sequence alignment for the initial target proteins [36]. Secondly, to expand the filtering scope, we constructed a new HMM model with e-value < 1 × 10^−20^. The new model was used to filter second target proteins using HMMER (version 3.3.2), with e-value < 0.05. The two results were combined and used as the final candidate proteins. Finally, the NCBI Conserved Domain Search (https://www.ncbi.nlm.nih.gov/Structure/bwrpsb/bwrpsb.cgi, accessed on 6 June 2022), Pfam Batch Sequence Search (http://pfam.xfam.org/search#tabview=tab1, accessed on 6 June 2022), and the SMART program (http://smart.embl.de/smart/batch.pl, accessed on 6 June 2022) [37] were used to verify the existence of the GRAS domain in each candidate protein sequence. After combining all results, 46 *GRAS* genes were obtained from the *T. cacao* genome.

### 2.2. Physicochemical Properties and Subcellular Localization Analyses of TcGRAS Genes

The online software ProtParam (https://web.expasy.org/protparam, accessed on 7 June 2022) and Compute pI/Mw (https://web.expasy.org/compute_pi, accessed on 7 June 2022) in the Expasy web server were used to analyze the physicochemical properties of the 46 GRAS proteins identified, including the theoretical isoelectric point (pI), molecular weight (MW), instability index, and aliphatic index [38]. Amino acid (aa) numbers and open reading frame (ORF) lengths were obtained on the ORFfinder website. (https://www.ncbi.nlm.nih.gov/orffinder, accessed on 7 June 2022). The subcellular localization (SL) of TcGRAS proteins was predicted by the BUSCA online program (https://busca.biocomp.unibo.it, accessed on 7 June 2022).

### 2.3. Phylogenetic Analysis and Classification of TcGRAS Genes

To provide family classification of GRAS genes and understand their phylogenetic relationships, a rooted neighbor-joining (NJ) phylogenetic tree between *T. cacao* (*TcGRAS*) and *Arabidopsis* GRAS proteins was built using the MEGA 11 software (version 11.0.11) [39,40]. The *TcGRAS* genes were classified according to their phylogenetic relationship with *A.thaliana* GRAS members. We obtained Arabidopsis GRAS protein sequences from TAIR (https://www.Arabidopsis.org, accessed on 10 June 2022) [9,41]. Both families of protein sequences were aligned by Muscle [42] in MEGA 11 software (version 11.0.11) under the default parameters. The maximum likelihood (ML) method was used with the following parameters: 1000 iterations for the bootstrap method, the Poisson model, and use all sites. In addition, an individual phylogenetic tree of *TcGRAS* genes was constructed in the same way and visualized using online software iTOL (http://itol.embl.de/, accessed on 10 June 2022) [43].

### 2.4. Gene Structure and Conserved Motif Analyses of TcGRAS Genes

The conserved motifs of the TcGRAS proteins were predicted by using the online program MEME (https://meme-suite.org/meme/tools/meme, accessed on 22 June 2022) with the following settings: maximum number of motifs 15, minimum motif width 6, maximum motif width 50, and any number of repetitions [44]. The domain analyses of TcGRAS proteins were performed under the Gene Structure Display Server 2.0 program. The gene structure view function of TBtools (version 0.665) was used to obtain conserved motifs and gene structures.

### 2.5. Chromosomal Mapping and Cis-Acting Regulatory Analyses of TcGRAS Genes

The online program MG2C (http://mg2c.iask.in/mg2c_v2.1, accessed on 15 June 2022) was used to predict the chromosomal position of *TcGRAS* genes. All the identified genes were mapped to 10 chromosomes according to the location information of the chromosome by TBtools. The upstream 2000 bp sequences of *TcGRAS* genes’ CDS were extracted by TBtools software (version 1.098696), and then submitted to the online software PlantCARE (http://bioinformatics.psb.ugent.be/webtools/plantcare/html, accessed on 15 June 2022) [45] to predict cis-acting elements, including light-responsive elements, abscisic acid-responsive elements, MeJA-responsive elements, low-temperature-responsive elements, defense- and stress-responsive elements, gibberellin-responsive elements, and auxin-responsive elements, after filtering and screening [45]. The Simple BioSequence Gene Viewer function of TBtools software (version 1.098696) was used to visualize the cis-acting elements.

### 2.6. Gene Duplication and Synteny Analyses of TcGRAS Genes

The ‘MCScanX’ function of the TBtools software with default parameters was used to predict gene duplications of *TcGRAS* genes. MCScanX Diamond output was used to calculate the replication events of the *T. cacao* genome. The Duplicate_gene_classifier program in MCScanX (https://github.com/wyp1125/MCScanX, accessed on 22 June 2022) was used to analyze the duplication type of each *TcGRAS* gene. KaKs_calculator software (version 2.0) [46] was used to calculate the Ka/Ks ratio of tandem repeat gene pairs in the *TcGRAS* gene, with the following parameters: method of calculation: YN, and genetic code Table 1 (Standard code). The Advanced Circos function of TBtools software (version 1.098696) was used to visualize WGD or segment duplications. The synteny of *TcGRAS* genes with the *GRAS* genes of *A. thaliana*, *P. trichocarpa*, and *S. indicum* was visualized by the One-Step MCScanX function of TBtools software. The Dual Systeny Plot for the MCScanX function of TBtools software (version 1.098696) was used to visualize the synteny.

### 2.7. GO Annotation Analyses of T. cacao TcGRAS Genes

The DAVID online program was used to annotate *TcGRAS* genes. The official gene sample lists of *TcGRAS* genes were uploaded to the program. The analysis included three parts: molecular function, cell components, and biological processes. The R programming language (version 4.1.3) was used to visualize the GO annotation analysis [47].

## 3. Results

### 3.1. Identification of GRAS Members in T. cacao

A total of 70 GRAS protein candidates were obtained from the initial filtering. After the second filtering, 53 candidate proteins were obtained. Finally, 46 *GRAS* genes were identified by redefining conserved domains and deleting repeats (Appendix A). The identified genes were named from *TcGRAS1* to *TcGRAS46* according to their chromosomal position. The number of amino acids (aa), average molecular weight (MW), theoretical pI, instability index, and aliphatic index of identified *TcGRAS* genes were statistically analyzed (Table 2). The number of amino acids of the *TcGRAS* genes ranged from 347 (TcGRAS21) to 1659 (TcGRAS22) aa, and the molecular weight ranged from 39,709.90 to 191,183.51 Da. The results showed that 44 GRAS proteins were acidic with pI values less than 6.5. Two (TcGRAS2 and TcGRAS19) were neutral, with pI between 6.5 and 7.5. The results of the instability index analysis showed that most TcGRAS proteins were unstable, except for TcGRAS8, TcGRAS12, TcGRAS24, and TcGRAS33. Prediction of the subcellular localization of TcGRAS proteins by the online software BUSCA tool revealed that 41 TcGRAS proteins were mainly located in the nucleus, 4 in the chloroplasts, and only 1 in the endomembrane system.

### 3.2. Phylogenetic Analysis of TcGRAS and AtGRAS

To explore the evolutionary relationship of GRAS proteins between *T. cacao* and *A. thaliana*, we performed a multiple sequence alignment of 46 TcGRAS proteins and 34 AtGRAS proteins, and then constructed an unrooted phylogenetic tree using the MEGA 11 software (Figure 1). According to the homology of GRAS proteins in *A. thaliana*, 46 TcGRAS proteins were divided into 10 clades, which were HAM, LISCL, LAS, SCL4/7, SCR, DLT, SCL3, DELLA, SHR, and PAT1. It is notable that 5 of the 46 TcGRAS proteins were not classified as any of these subfamilies; therefore, we grouped TcGRAS22 as UN1 and TcGRAS11, TcGRAS13, TcGRAS17, and TcGRAS33 as UN2. The largest clade was subgroup LISCL, which contained ten TcGRAS members (TcGRAS8, TcGRAS24, TcGRAS25, TcGRAS27, TcGRAS39, TcGRAS40, TcGRAS41, TcGRAS42, TcGRAS43, and TcGRAS46), whereas subgroups UN1, DLT, SCL4/7, and LAS only had one member. Subgroups UN1 and UN2 only contained *T. cacao* members, meaning that these genes may have been specialized during the evolutionary process.

### 3.3. Gene Structure, Conserved Motifs, and Domain Analyses of TcGRAS Genes

To understand the structural diversity and similarity of GRAS gene family members in the cacao tree, we used the Gene Structure View function of the TBtools software to construct a triad map of the evolutionary tree, gene structure, and motif of GRAS gene family members, as shown in Figure 2. We first performed an individual phylogenetic tree using an NJ method consistent with the phylogenetic analysis between TcGRAS and AtGRAS (Figure 2A).

To further understand the characteristics of the GRAS gene families in *T. cacao* and the conserved motifs shared among different subfamilies, we used the Multiple Expectation Maximization for Motif Elicitation program to find the conserved motifs. A total of 10 conserved motifs were predicted and named Motif 1–10 (Figure 2B and Appendix A). Early sequence analysis indicated that the GRAS proteins typically share a variable N terminus and a highly conserved C terminus. In this study, we found that the C-terminal regions contained a highly conserved domain (Motif 6). Three proteins did not contain this conserved motif, including TcGRAS2, TcGRAS20, and TcGRAS21, and we hypothesized that the C-terminal region of these GRAS proteins was truncated, lacking part of the GRAS domain.

We used the Gene Structure Display Server 2.0 program to construct a domain analysis of TcGRAS proteins (Figure 2C). We found a total of eight types of conserved domains. All the *GRAS* genes contain the GRAS domain or GRAS superfamily domain. In addition, the domains of GRAS members in DELLA also have the DELLA superfamily. The *TcGRAS2* gene has a GRAS superfamily and TB2_DP1_HVA22 superfamily domain. The *TcGRAS22* gene has a GRAS superfamily, ZnF_BED, DUF4413, Dimer_Tnp_hAT, and Peptidase_c48 superfamily domain.

### 3.4. Chromosomal Mapping and Cis-Acting Regulatory Analyses of TcGRAS Genes

The location of *TcGRAS* genes was obtained from genome annotation files. A total of 46 *TcGRAS* genes were randomly distributed on 10 chromosomes and were named from TcGRAS1 to TcGRAS46 according to their positions on the chromosomes (Figure 3A). Chr01 and Chr04 had the largest number (nine, 19.57%) of *TcGRAS* genes, followed by Chr09 with eight members (17.39%). In contrast, Chr05, Chr06, and Chr10 contained only two *TcGRAS* genes each (4.35%). Chr04 contained seven subgroups of *TcGRAS* genes, as shown in Figure 3B, while Chr05, Chr06, and Chr10 contained only two subgroups each. Subgroup DLT was only observed on Chr03, subgroup SCL4/7 was only observed on Chr01, and subgroup LAS was only observed on Chr07.

Cis-acting elements regulate transcription initiation and transcription activity by binding to transcription factors. To explore the promoter function of *TcGRAS* genes, we extracted 2000 bp sequences upstream of the transcription start site. Then, we submitted these to the online program PlantCARE. Seven types of important cis-acting elements were obtained after sorting and screening, including light-responsive element, abscisic acid-responsive element, MeJA-responsive element, defense and stress-responsive element, low-temperature-responsive element, auxin-responsive element and gibberellin-responsive element. The light-responsive element was found in all promoter regions of *TcGRAS* genes. In addition, more than half of the 46 *GRAS* genes had the abscisic acid-responsive element, MeJA-responsive element, and gibberellin-responsive element. Compared with the MADS-Box transcription factor family in cacao tree, the *GRAS* gene family contains significantly more light-responsive elements, abscisic acid-responsive elements, and MeJA-responsive elements. The distribution of these cis-acting elements is shown in Appendix A.

### 3.5. Gene Duplication and Syntenic Analysis of TcGRAS Genes

Genome-wide analysis of cacao tree gene replication by MCScanX software revealed 2148 tandem duplicated genes in the cacao tree genome, while only 2 pairs of tandem duplicated genes were present among 46 *TcGRAS* genes (Figure 4). The analysis showed that one pair of tandem duplication genes (*TcGRAS24* and *TcGRAS25*) was located on Chr04, and another pair (*TcGRAS42* and *TcGRAS43*) on Chr09. In addition, the substitution ratio of non-synonymous (Ka) to synonymous (Ks) mutations (Ka/Ks) of two pairs were calculated (Table 3). The Ka/Ks values of both pairs were more than 1, indicating that these genes were positively selected over the course of evolution, and the novel protein functions may be beneficial to the survival and reproduction of *T. cacao*.

The MCScanX showed that there were 2767 segmental duplications in the genome of *T. cacao*, and only 6 pairs of fragmental duplicated genes were predicted out of 46 identified *TcGRAS* genes. The Advanced Circos function of the TBtools software was used to visualize the segmental duplication of *GRAS* genes on 10 chromosomes, as shown in Figure 4. Chr01 contained three duplicated genes, and Chr02, Chr03, and Chr04 contained two duplicated genes, while Chr05, Chr06, and Chr09 each contained only one duplicated gene. However, Chr07, Chr08, and Chr10 did not contain any segmental duplicated genes.

The syntenic analyses of *TcGRAS* genes with the *GRAS* genes of *A. thaliana*, *P. trichocarpa*, and *S. indicum* were separately analyzed to find homologous gene pairs (Figure 5). A total of 36 *GRAS* genes of *T. cacao* had a syntenic relationship with the *GRAS* genes of *A. thaliana* (16), *P. trichocarpa* (32), and *S. indicum* (30). Some genes had multiple syntenic relationships with other closely related species. Therefore, a total number of 25 (Appendix A), 77 (Appendix A), and 48 (Appendix A) *GRAS* genes of *A. thaliana*, *P. trichocarpa*, and *S. indicum*, respectively, had synteny with 36 *GRAS* genes. Furthermore, it was found that 13 *GRAS* genes existed in these 4 plants at the same time (Figure 6). Two homologous *GRAS* genes existed in *T. cacao* and *S. indicum* rather than in *P. trichocarpa* and *A. thaliana*. Similarly, *T. cacao*, *P. trichocarpa,* and *S. indicum* had 13 homologous *TcGRAS* genes that did not exist in *A. thaliana*, *T. cacao*, *P. trichocarpa,* and *A. thaliana* had 1 homologous *TcGRAS* gene that did not exist in *S. indicum,* and *T. cacao*, *S. indicum,* and *A. thaliana* had 2 homologous *GRAS* genes that did not exist in *P. trichocarpa*. Five homologous *GRAS* genes existed in *T. cacao* and *P. trichocarpa* but did not exist in *A. thaliana* and *S. indicum*.

### 3.6. GO Annotation of T.cacao TcGRAS Proteins

To understand TcGRAS protein function in different biological processes, we performed a GO annotation analysis of the *TcGRAS* genes (Figure 7), and the GO numbers are shown in Appendix A. The analysis of the cellular composition showed that most of the TcGRAS proteins were mainly concentrated in the nucleus. The analysis of biological processes showed that the *TcGRAS* genes were involved in many biological processes. The large portions of GRAS proteins were involved in transcriptional regulation. Otherwise, some TcGRAS proteins were involved in the negative regulation of biological processes, for example, negative regulation of seed germination and the gibberellic acid-mediated signaling pathway. In addition, some *TcGRAS* genes also respond to abiotic stresses and regulate plant organ development. The analysis of the molecular functions of *TcGRAS* genes revealed that they had functions in transcription factor activity and sequence-specific DNA binding.

## 4. Discussion

In this study, we provided the first comprehensive analysis of the *GRAS* gene family in *T. cacao*. Based on the latest genome sequences and annotation files, we identified 46 *GRAS* genes distributed across 10 chromosomes in the *T. cacao* genome. These 46 *TcGRAS* genes were classified into 12 subgroups (HAM, LISCL, LAS, SCL4/7, SCR, DLT, SCL3, DELLA, SHR, PAT1, UN1, and UN2) according to their phylogenetic relationship with *A. thaliana*. We found that the GRAS gene family members were unevenly distributed among subgroups; for instance, the subgroups of UN1 and UN2 only contained *T. cacao* members, and the member number of subgroups of SCR and SCL3 in *T. cacao* was more than that in *A. thaliana*. During the evolution of gene families, the gene structure changes in response to environmental changes to acquire new functions. The structural analysis of *TcGRAS* genes according to phylogenetic relationships showed that different subgroups had different gene structures and conserved motifs, while the same subgroup had similar motifs and gene structures, which meant that members of the same subgroup had similar functions. Since *T. cacao* and *A. thaliana* were exposed to different environments during their evolutionary processes, the number of *GRAS* genes in their subgroups became different as GRAS genes differentiated.

By analyzing the intron/exon structure of the *TcGRAS* genes, we found that majority of these genes were free of introns, which was similar to the observed lack of introns in Arabidopsis and rice *GRAS* genes [8]. A previous study showed that ancestors of each eukaryote had intron-rich genes and that extensive loss and insertion of introns from most genes may have occurred due to selective pressure, with gene duplication accelerating this process [48,49]. Nevertheless, some *GRAS* genes have evolved different intron/exon structures, indicating that they likely evolved new specialized functions to adapt to their environment.

Tandem and segmental duplications are thought to be the main mechanisms contributing to the expansions of gene families in plants [50]. Both tandemly and segmentally duplicated genes that have been retained in plant genomes play important roles in adaptive responses to environmental stimuli [51,52]. The collinearity analysis in our study showed that there were two pairs of tandem duplication and six pairs of segmental duplication events in the *T. cacao* GRAS gene family, and this might play an important role in the GRAS family expansion in *T. cacao*.

The cis-acting elements play a vital role in regulating gene expression during plant growth and development [53]. The promoter analysis showed that the light-responsive element was found in all promoter regions of *TcGRAS* genes. In addition, more than half of the 46 *GRAS* genes had the abscisic acid-responsive element, MeJA-responsive element, and gibberellin-responsive element, which made it possible to study the function of these genes in the future.

To further analyze the function of GRAS transcription factors in *T. cacao*, we studied the end of the genotype affected by functional diversity after GO enrichment analysis, and the results showed that the majority of cacao GRAS proteins play an important role in many different biological processes, including abiotic stresses and plant organ development.

## 5. Conclusions

In this study, we identified and systematically analyzed the GRAS gene family in *T. cacao*. Based on the genomic data of the cacao tree, we finally identified 46 GRAS genes using double HMM profiles. These 46 *GRAS* genes were distributed on 10 chromosomes and phylogenetically divided into 12 subfamilies, with highly similar gene structures and conserved motifs within the same subfamily. Cis-acting element analysis indicated that *GRAS* genes may be involved in various abiotic stress responses. In addition, we found that tandem and segmental duplications contribute to the expansions of the GRAS gene family. A further syntenic analysis showed that the functions of *TcGRAS* genes might be speculated from the function of *GRAS* genes in other plants. Through GO analysis, we found that most of the *TcGRAS* genes were involved in transcriptional regulation. In summary, the results provide information for further research of the TcGRAS genes’ function and lay the foundation for further investigation.

## Figures and Tables

**Figure 1 genes-14-00057-f001:**
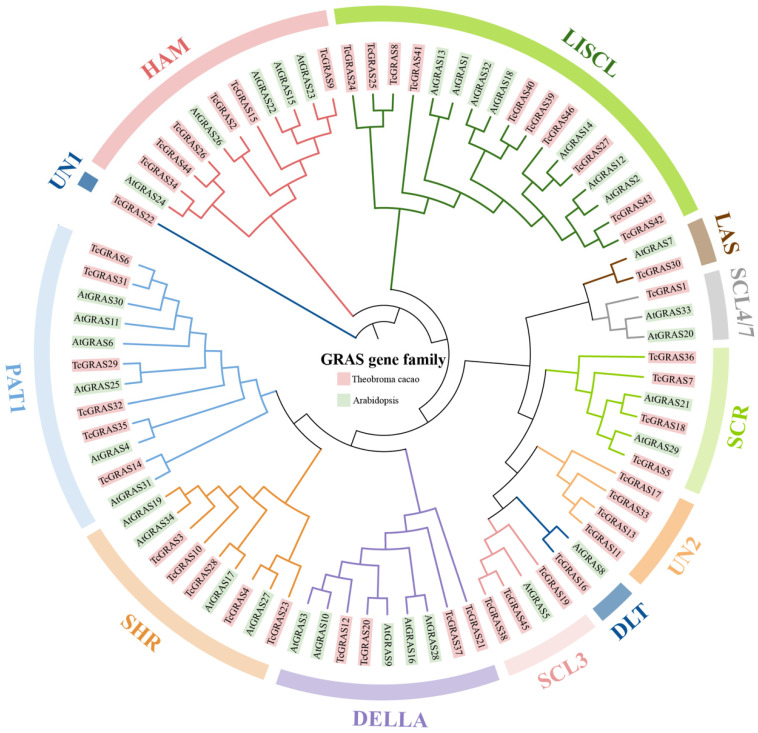
Phylogenetic tree of GRAS genes in *T. cacao* and *A. thaliana*. The GRAS genes in *T. cacao* and *A. thaliana* are shown in red and green, respectively. The tree branched the GRAS proteins into different subgroups, illustrated by different colored clusters within the clade. TcGRAS are divided into twelve subgroups according to the subgrouping of *A. thaliana*. The phylogenetic tree was constructed using the maximum likelihood (ML) method with 1000 bootstrap replications of the MEGA 11 software.

**Figure 2 genes-14-00057-f002:**
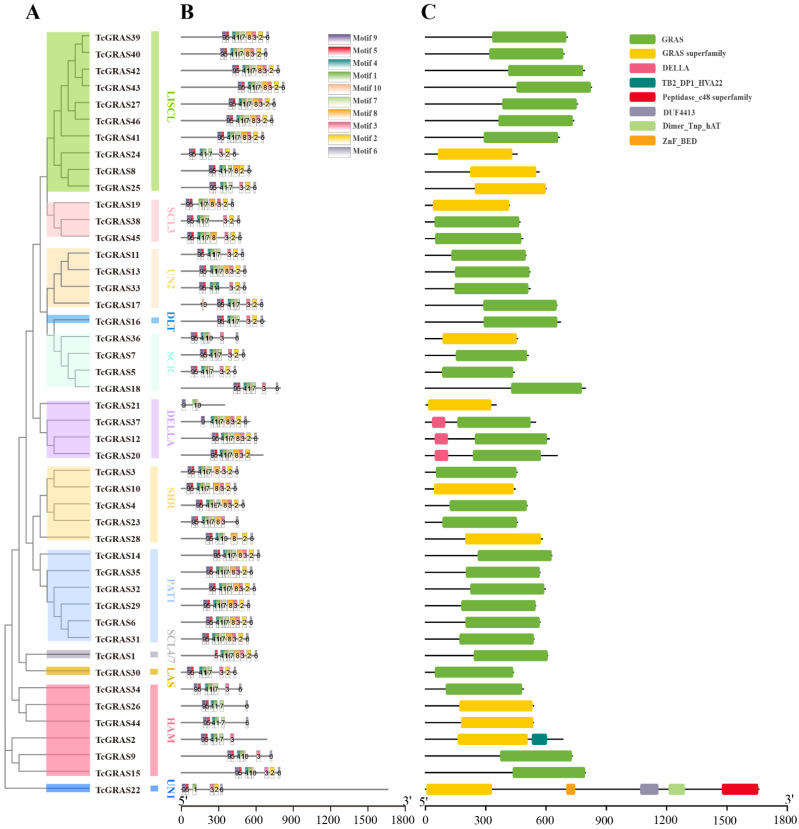
Phylogenetic relationship, conserved motifs, and gene structure of *TcGRAS* genes. (**A**) An unrooted phylogenetic tree of TcGRAS proteins was constructed using MEGA 11 software under the ML method with 1000 bootstrap replicates. (**B**) Conserved TcGRAS proteins’ motifs performed by MEME. The colored boxes refer to motifs. The black lines refer to non-conserved sequences. The scale bar is 300 amino acids. (**C**) The domain analyses of TcGRAS proteins were performed under the Gene Structure Display Server 2.0 program. The domains are displayed in different colored boxes.

**Figure 3 genes-14-00057-f003:**
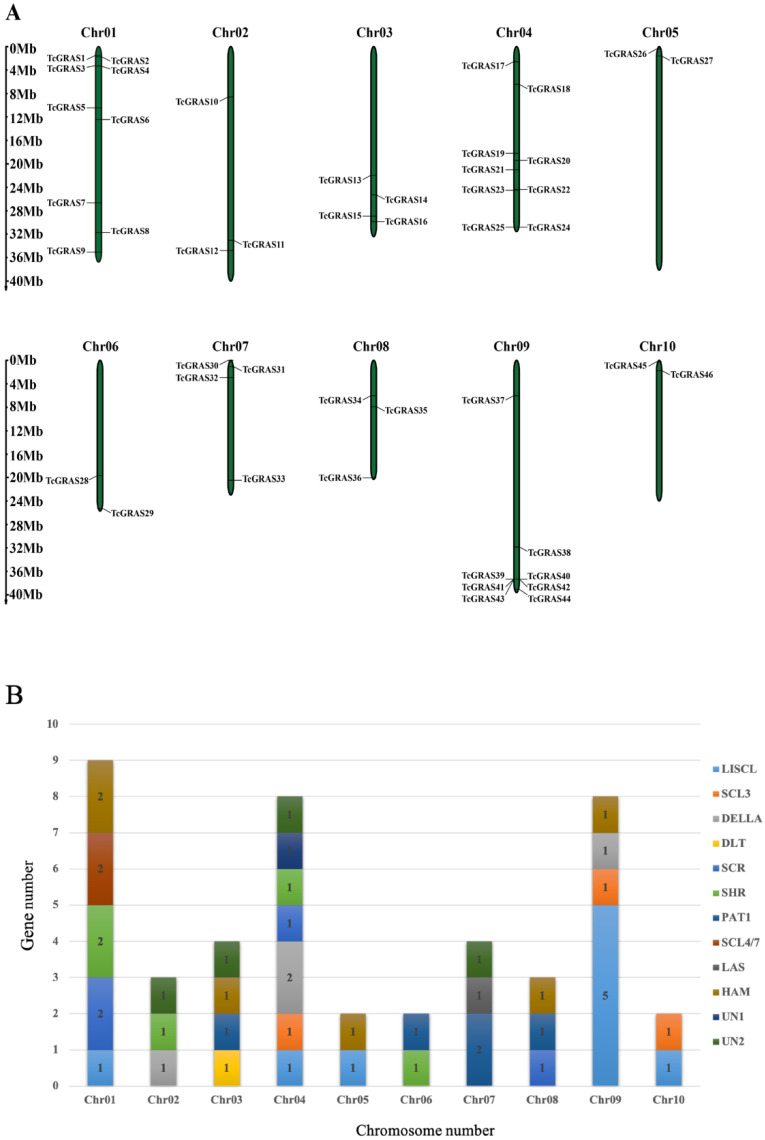
(**A**) Physical distribution of *TcGRAS* genes among 10 chromosomes. (**B**) Number of TcGRAS subfamilies on each chromosome.

**Figure 4 genes-14-00057-f004:**
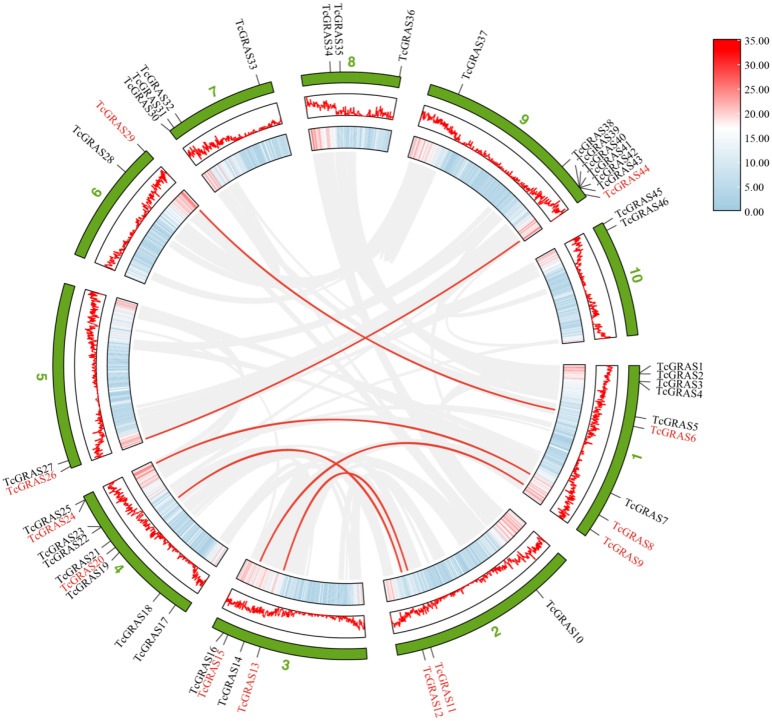
Schematic diagram of the duplication patterns of the *TcGRAS* genes. The red lines show segmental duplications of TcGRAS gene pairs. The gray lines show segmental duplications of all gene pairs in the *T. cacao* genome. The first ring from outside indicates the chromosomal localization of 46 putative GRAS genes in *T. cacao*. The second and third rings from outside represent the density of genes on the chromosomes. The blue-to-red scale bar on the right indicates the number of SNPs within 1 Mb window size.

**Figure 5 genes-14-00057-f005:**
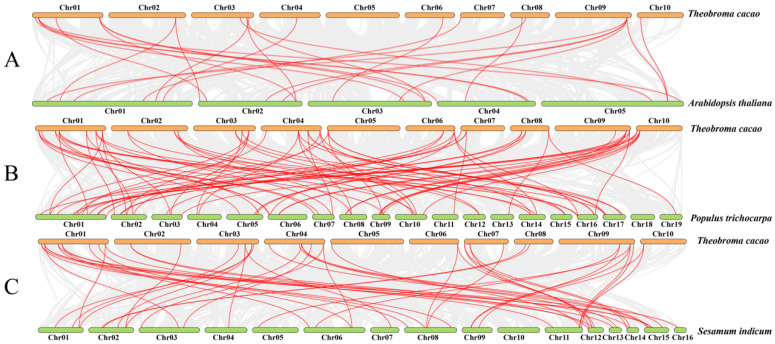
Visualization of the syntenic analysis. Synteny of the *GRAS* genes in *T. cacao* with the *GRAS* genes of *A. thaliana* (**A**), *T. cacao* (**B**), and *S. indicum* (**C**) was visualized by MCScanX analysis of TBtools software. Gray lines between the genomes show all synteny blocks, and red lines between the genomes indicate the synteny between the genes.

**Figure 6 genes-14-00057-f006:**
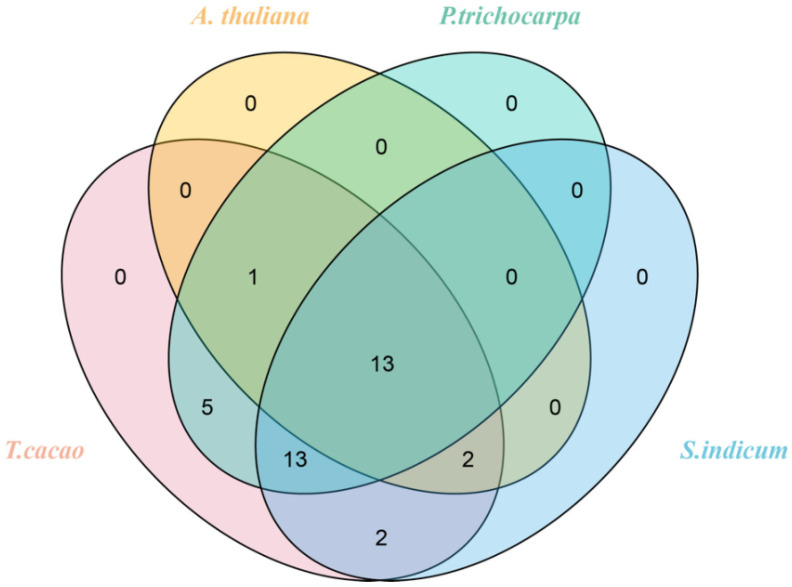
Venn diagram of the identical and different *GRAS* genes among *T. cacao*, *A. thaliana*, *P. trichocarpa*, and *S. indicum*.

**Figure 7 genes-14-00057-f007:**
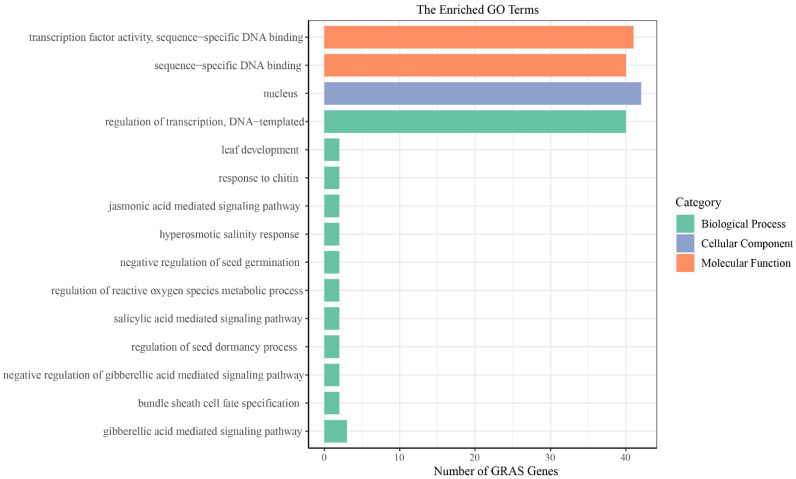
GO annotation of TcGRAS proteins. Green represents biological processes, purple represents cellular components, and orange represents molecular functions.

**Table 1 genes-14-00057-t001:** Recent advances in biological functions of the GRAS transcription factor gene family.

Gene	Subfamily	Function	Species	Classified Subfamilies	Reference
*HAM*	HAM	Maintenance of stems	*Petunia hybrida*	unclassified	[17]
*LISCL*	LISCL	Regulating the transcription process during microsporogenesis	*Lilium longiflorum*	unclassified	[18]
*Ls*	LAS	Initiation of the axillary meristem	*tomato*	HAM, LAS, SCL4/7, SCR, SCL9, SCL28, DELLA, SHR, PAT1, Os4, Os19, GRAS37, and Pt20	[19]
*PeSCL7*	SCL4/7	Enhanced drought tolerance and salt tolerance of transgenic Arabidopsis plants	*Populus euphratica*	unclassified	[20]
*AtSCR*	SCR	Involved in radial root morphology and growth	*A. thaliana*	HAM, LISCL, LAS, SCL4/7, SCR, DLT, SCL3, DELLA, SHR, and PAT1	[4]
*AtSHR*	SHR	Involved in radial root morphology and growth	*A. thaliana*	HAM, LISCL, LAS, SCL4/7, SCR, DLT, SCL3, DELLA, SHR, and PAT1	[21]
*PtSHR1*	SHR	Increased growth rates	*Populus tomentosa*	Os19, HAM, Os4, Pt20, DLT, AtSCl3, AtSHR, AtPAT1, AtSCR, AtSCL4/7, AtLAS, DELLA, and LISCL	[22]
*DLT*	DLT	Involved in brassinoltone signaling	*rice*	LISCL, SHR, DELLA, SCL3, PAT1, SCR, SCL4/7, LAS, Os19, HAM, Os4, and DLT	[23]
*AtSCL3*	SCL3	Integrated multiple signals during Arabidopsis root cell elongation	*A. thaliana*	HAM, LISCL, LAS, SCL4/7, SCR, DLT, SCL3, DELLA, SHR, and PAT1	[24]
*AtRGA*	DELLA	Modulated jasmonate signaling via competitive binding to JAZs	*A. thaliana*	HAM, LISCL, LAS, SCL4/7, SCR, DLT, SCL3, DELLA, SHR, and PAT1	[25]
*BdSLR1 BdSLRL1*	DELLA	Play a role in plant growth via the GA signal pathway	*Brachypodium distachyon*	HAM, PAT1, SHR, DELLA, SCL3, SCL4/7, LAS, SCR, DLT, and LISCL	[12]
*AtPAT1*	PAT1	Involved in signaling in Arabidopsis photochromes	*A. thaliana*	HAM, LISCL, LAS, SCL4/7, SCR, DLT, SCL3, DELLA, SHR, and PAT1	[26]
*GmGRAS37*	PAT1	Improved resistance to drought and salt stresses	*soybean*	AtSCL4/7, Os19, Os4, HAM, DELLA, DLT, AtPAT1, LISCL, AtSCR, AtSCL3, and AtSHR	[10]
*StGRAS9*	PAT1	Responded to plant hormones IAA, ABA, and GA3 treatment	*potato*	DELLA, LAS, HAM, PATI, SCR, LISCL, SHR, and SCL3	[13]

**Table 2 genes-14-00057-t002:** Physicochemical properties and subcellular localization analyses of the GRAS gene family in *T. cacao*.

Gene Name	Gene ID	Physicochemical Characteristics					SL	ORF
		PI	MW (Da)	Length (aa)	Instability Index	Aliphatic Index		
*TcGRAS1*	TCM_000399	5.00	67,286.55	608	54.24	81.73	endomembrane system	1827
*TcGRAS2*	TCM_000435	6.80	77,476.49	684	50.67	84.71	nucleus	2055
*TcGRAS3*	TCM_000764	5.46	50,905.28	456	41.68	92.43	chloroplast	1371
*TcGRAS4*	TCM_000801	5.59	57,001.87	505	52.26	70.32	chloroplast	1518
*TcGRAS5*	TCM_002021	5.42	48,046.37	441	56.70	91.54	nucleus	1326
*TcGRAS6*	TCM_002319	5.57	63,858.30	569	55.85	79.17	nucleus	1710
*TcGRAS7*	TCM_003984	5.84	57,336.13	511	44.83	83.99	nucleus	1536
*TcGRAS8*	TCM_004818	6.15	63,796.67	565	31.39	85.91	nucleus	1698
*TcGRAS9*	TCM_005571	5.41	79,449.91	730	62.24	84.47	nucleus	2193
*TcGRAS10*	TCM_007806	5.31	49,737.63	445	43.06	88.97	chloroplast	1338
*TcGRAS11*	TCM_010708	6.38	56,277.83	499	47.40	92.06	nucleus	1500
*TcGRAS12*	TCM_010965	5.66	67,186.67	615	39.21	80.93	nucleus	1848
*TcGRAS13*	TCM_014574	6.04	57,510.78	519	52.58	90.02	nucleus	1560
*TcGRAS14*	TCM_015228	5.76	67,975.59	628	45.58	77.82	nucleus	1887
*TcGRAS15*	TCM_015991	5.69	87,113.44	795	55.54	80.34	nucleus	2388
*TcGRAS16*	TCM_016186	5.68	74,493.64	671	52.33	79.79	nucleus	2016
*TcGRAS17*	TCM_017269	5.67	72,460.97	654	53.85	78.35	nucleus	1965
*TcGRAS18*	TCM_017746	5.88	86,321.23	795	53.09	81.66	nucleus	2388
*TcGRAS19*	TCM_018964	6.92	46,686.93	418	44.25	94.07	nucleus	1257
*TcGRAS20*	TCM_019165	5.27	71,951.43	655	49.47	81.62	nucleus	1968
*TcGRAS21*	TCM_019414	5.43	39,709.90	347	49.00	94.96	nucleus	1044
*TcGRAS22*	TCM_019956	6.08	191,183.51	1659	47.01	80.49	nucleus	4980
*TcGRAS23*	TCM_019978	5.82	51,125.89	457	46.55	79.63	chloroplast	1374
*TcGRAS24*	TCM_021350	5.67	50,849.34	457	33.62	93.09	nucleus	1374
*TcGRAS25*	TCM_021351	5.08	67,410.79	600	47.40	83.42	nucleus	1803
*TcGRAS26*	TCM_021618	4.86	61,003.74	538	49.81	70.91	nucleus	1617
*TcGRAS27*	TCM_021920	6.17	85,337.10	755	52.92	73.09	nucleus	2268
*TcGRAS28*	TCM_029136	6.30	64,717.65	582	46.26	74.60	nucleus	1749
*TcGRAS29*	TCM_030393	5.89	61,290.47	548	52.83	79.00	nucleus	1647
*TcGRAS30*	TCM_030498	6.19	49,455.27	438	55.76	96.74	nucleus	1317
*TcGRAS31*	TCM_030733	5.56	60,140.03	540	58.44	79.35	nucleus	1623
*TcGRAS32*	TCM_031132	6.13	66,239.71	596	52.40	80.54	nucleus	1791
*TcGRAS33*	TCM_033446	4.94	58,582.29	521	39.07	86.91	nucleus	1566
*TcGRAS34*	TCM_035069	5.28	53,980.68	487	40.15	85.15	nucleus	1464
*TcGRAS35*	TCM_035362	5.06	63,856.24	570	49.25	82.14	nucleus	1713
*TcGRAS36*	TCM_036707	6.31	52,208.63	457	43.57	86.81	nucleus	1374
*TcGRAS37*	TCM_037975	5.62	60,414.53	548	49.30	87.08	nucleus	1647
*TcGRAS38*	TCM_040833	5.61	52,722.63	470	50.42	98.81	nucleus	1413
*TcGRAS39*	TCM_041810	6.33	79,445.42	705	46.24	71.65	nucleus	2118
*TcGRAS40*	TCM_041812	6.17	78,589.84	690	51.06	80.39	nucleus	2073
*TcGRAS41*	TCM_041813	5.40	75,260.51	666	53.55	81.14	nucleus	2001
*TcGRAS42*	TCM_041814	5.16	89,125.20	790	45.83	71.96	nucleus	2373
*TcGRAS43*	TCM_041815	6.30	92,905.70	829	46.81	70.95	nucleus	2490
*TcGRAS44*	TCM_042194	4.63	60,211.98	537	45.66	85.74	nucleus	1614
*TcGRAS45*	TCM_042392	5.80	54,285.57	483	62.33	96.15	nucleus	1452
*TcGRAS46*	TCM_042705	5.62	83,092.65	737	52.65	76.73	nucleus	2214

**Table 3 genes-14-00057-t003:** Tandem duplication in *TcGRAS* genes and corresponding Ka, Ks, and Ka/Ks values.

Tandem Duplication	Chromosome Name	Ka	Ks	Ka/Ks
TcGRAS24 and TcGRAS25	Chr04	5.21	2.66	1.96
TcGRAS42 and TcGRAS43	Chr09	5.62	1.41	3.99

## Data Availability

Data are contained within the article or Appendix A.

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
