# Peer review of "Genome-Wide Identification and Analysis of the GRAS Transcription Factor Gene Family in *Theobroma cacao"

_genes, 2022, doi:10.3390/genes14010057_

Round 1
Reviewer 1 Report
This manuscript reports the results of a computational analysis of the GRAS family of transcription factors in cocoa (Theobroma cacao). A total of 46 genes are identified and sorted into 12 subfamilies based on their relatedness to GRAS proteins in Arabidopsis. Many GRAS factors play important roles in other plants, and researchers studying cocoa may find this a useful survey of the GRAS landscape in their favorite plant.
The abstract creates the impression that experiments were performed, even though the actual analysis was entirely computational. The description in the introduction of all the GRAS gene subfamilies in different plants may be more digestible if presented in a table format. A more thorough description of what all the parts of Figure 4 are showing would be helpful. Does it make sense for a transcription factor to be localized in a membrane or the extracellular space? Is the number transcription factor binding sites identified in the promoters of GRAS genes higher or lower than for other genes in the genome?
Is the genome annotation correct on which structure of the genes shown in Figure 2C is based? Do nine of the GRAS genes really contain a 5’-UTR that is longer than 3,000 nt, four of which appear longer than 6,000 nt? It is confusing because the legend to Figure 2 says the green boxes are exons but the inset depicts them as UTRs. Does TcGRAS21 really have an intron that is roughly 12,000 nt long? For that matter, is TcGRAS21 even a legitimate GRAS gene if it only contains 3 of the defining motifs that all other GRAS genes have at least 6 of? This could be clarified if the criteria through which GRAS genes were identified was described more explicitly.
The purpose of the synteny analysis was not clear to me. Figure 5 shows that the genomes of these distantly related plants are quite thoroughly scrambled relative to each other. Figure 6 is misleading in that it gives the incorrect impression that there are 0 GRAS genes in common between Arabidopsis, Populus, and Sesamum except those that are present in cocoa.
Many of the conclusions drawn are simplistic to the point of being obvious, such as “gene duplication might be related to GRAS gene family amplification”, “these genes may have been differentiated during the evolutionary process”, and “Remarkably, TcGRAS genes in the same subfamily had similar motifs, we speculated that they may have similar biological functions.”
The manuscript is riddled with errors. In the second paragraph of the introduction (lines 41-43), the GRAS domain is described as containing five units but only 4 are listed. Populus is said to contain 11 subfamilies of GRAS proteins but 13 are listed on lines 55-56. TcGRAS21 is described on line 236 as being the shortest GRAS gene at 1044 bp in length but Figure 2C shows it is one of the two longest GRAS genes at around 15,000 bp. The UN2 group is described in line 230 and again in line 344 as containing multiple introns but Figure 2C shows it is group UN1 that has all the introns. The references to Figures 2B and 2C are reversed in the text. Reference 49 shows the opposite of what it is cited to support. And so on and so on. This all creates the impression of a manuscript prepared in haste.
Reviewer 2 Report
The manuscript performed a genome wide comprehensive in silico analysis of the GRAS family in Theobroma cacao. The authors identified 46 GRAS genes in T. cacao genome, performed physiochemical properties, phylogeny, gene structure, chromosomal location and synthetic relationships.
The manuscript is average and requires polishing by a native speaker. Additionally, the below comments, observations and suggested corrections;
Between line 11 and 12, the study plant needs to be mentioned, the study species needs to be introduced.
Line 13.."analyses and performed are redundant". Simply use 'performed or 'analyses on physiochemical properties, phylogenetic relationships...
check grammar in line 11 and throughout the write up.
For a good flow of the write up, presentation of methods, results and discussion should be systematic and in tandem.
the genes distributed along the ten chromosomes were divided into 12 subfamilies. explain the basis of the subfamilies.
some sentences are wordy and redundant e.g in line 18 and 20, the authors could simply write it as ".....gene duplication analysis showed that ....delete.."The results.....
"related t" in line 22 is not appropriate, consider "account for"
Results on physiochemical properties, conserved sequences and phylogenetic relationships are missing from the abstract.
line 37 "the GRS transcription factors gene family is named....
second paragraph should start at line 37.
most phrases are rushed. for example, the authors need to describe the regions in the GRAs domain then preced to describe each of the mentioned regions. as it is, the re4ader questions where these regions arise fromand how many regions the GRAs domain has
Line 47 "Soybean is not a scientific name, why is it italicized? Tomato in line 49 as well.
line 49-52, rephrase to avoid starting sentences with numerals. the authors could start with the species e.g "in tomato....GRAs subfamilies were..."
Grammar: Line 68"....the initial...." did the authors mean "initiation"?
most of the species reviewed in line 47-58 are flowering plants.
Line 86: why are the authors sighting a need to study cocoa tree yet their study in molecular and not morphological aspects.
document writing structure, word choice and tone sounds more of narration that it is scientific.
in the introduction, the authors study is not well grounded in terms of background of the study. The authors need to review on the findings of other works with regards to GRAs aspects that they analysed.
MATERIALS AND METHODS
The research is predominantly in silico. The bioinformatics instruments are adequate and up to date. However, sentence structure needs revision.
Authors should be guided by the international nomenclature code when writing scientific names, e.g "Arabidopsis Thaliana " Line 39
Gene structure and conserved motif analyses of TcGRAs genes:
To describe the number of replications as "any" is ambiguous, the methodology should be reproducible
For the intro-exon structures, how was the gtf file generated? Authors should vividly explain their methodology for reproducibility and credibility of the presented results.
Line 153, (version 0.665), seems wrongly placed. Is this the version of gene structure view or the TB tools?
line 155 is hanging and is grammatically improper
After predicting the chromosomal position of the GRAs, which tools was used for mapping the identified GRAs genes onto the chromosomes?
2.6 Gene duplication and synteny analyses of TcGRAs genes
in the parameters used to calculate the ka/ks ratio of tandem repeat gene pairs, the authors should just refer to the table than opt to list the parameters and only provide one parameter.
Check sentence 175-177
RESULTS
Line 92, report as "the amino acids sequences of GRAs proteins ranged from ... to ...."
reporting protein sequences and amino acids is misleading...
Table 1: Title should be reflective of what is contained therein. Simply saying 'detailed information is vague"
Phylogenetic analysis
Line 203-206 is too lengthy, the authors should fragment it
what was the basis of the UNI1 & UNI2 classification since the rest were classified based on their homology to Arabidopsis GRAs.
213 "differentiated" what do the authors mean?
Figure 1 is not clear
Figure 1 caption is not clear.
An addition of GO annotation for predicted functional analysis will enrich the manuscript. In fact, the authors have reviewed on the functions of the GRAs in other species in their introduction.
MS however lacks adequate review of other researcher's findings on the parameters that they studied on, example is the physiochemical properties, motifs and location.
Gene structure, conserved motifs and domain analyses of TcGRAs
What was the length of the longest and shortest intron & exons
Which group had the longest exons/introns
Line 245: Any literature that supports the argument?
Where is the results on domain analysis?
It also notable that results have been presented as materials an methods. The authors should only present the findings in results section.
Discussion
Too brief . Not all results have ben discussed.
Additionally, the authors failed to compare their findings with previous findings, despite their use of GRAs family genes from other species.
line 351, The study has already been conducted yet the authors still speculative.
Line why didn't the study perform any functional validation either transcript expression levels or transformation (transient expression for instance)?
The manuscript requires typesetting? several sentences are starting with capital letters.
Additional comments have been highlighted in the pdf manuscript document attached.

Round 2
Reviewer 1 Report
This is a revised version of a manuscript I previously reviewed as Reviewer 1. Many of the changes I suggested were accepted and the manuscript is improved. In their response, the authors separated my original comments into 9 points and I will continue their system below.
Point 1. The abstract still creates the impression that experiments were done. This could be fixed with a few words making it clearer that the analysis was entirely computational, and that the cis-regulation of these genes and the subcellular localization of their products is predicted, not tested. The first, second, and fourth sentence of the new abstract seem better suited for the introduction.
Point 2. The new Table 1 is an improvement. It could be made even better by including the information from lines 59-70 showing the subfamilies of GRAS genes in different plants.
Point 3. The description of Figure 4 is better. What are the units on the red-to-blue scale bar that goes from 0 to 35?
Points 4, 5, and 6. What I was trying to do in my original review was to raise questions that illustrate the point that the authors should be more skeptical about the output of predictive programs and even the annotation of the published genome sequence. No predictive program or genome annotation is 100% accurate. Some critical analysis is needed to help identify likely mistakes, even in the absence of experimental evidence. Because a membrane or extracellular location seems incompatible with transcription factor function, some explanation is needed if this prediction is accepted as being correct. The output of predictive programs are hypotheses, not facts.
Point 5. The presence of a transcription factor binding site does not by itself mean that a gene is regulated by that transcription factor. The description in the manuscript of the frequency at which various elements are found in the promoters of GRAS genes is fine, but these results need some context or comparison to be meaningful. Do these elements show up more or less often in GRAS genes than in the promoters of other families of transcription factors? How about in families of genes that are not transcription factors?
Point 6. Because translation usually starts at the first AUG in a mRNA, 5’-UTRs longer than a few hundred nucleotides are very rare. Do the 5’-UTRs longer than 1 kb shown in Figure 2 lack AUGs? If not, some explanation is needed for how translation could initiate at the start of the GRAS coding sequences rather than all other instances of AUG closer to the 5’ end of the mRNA. A mistake in genome annotation could be one explanation.
Point 7. The synteny analysis is still very confusing. I don’t understand why only 36 of the 46 cacao GRAS genes they identified were included in Figure 6.
Point 8. Some overly simplistic conclusions were improved.
Point 9. The errors I pointed out last time were fixed or deleted.
Reviewer 2 Report
Thank you for the opportunity to re-review this manuscript. Having gone through the revised manuscript and subsequently the report, I must admit that the revisions made by the authors are not adequate to warrant publication of the paper in GENES;
1. The manuscript still has serious english issues. 2. The authors have not satisfactorily responded to the issues raised. 3. The content is inadequate, in silico analysis alone is not sufficient. And even after suggesting that the authors add GO enrichment/ontology(an in silico analysis), the authors failed to do so. Regardless, the manuscript requires at least a functional validation experiment which they insist can be done later.Author Response
Please see the attachment.

Round 3
Reviewer 1 Report
The authors have incorporated virtually all of the corrections and suggestions made in previous reviews. The paper is much easier to read and makes it clear that the work was entirely computational. The only obvious error I found is the repeated sentence from line 22 to 25 in the abstract.
Reviewer 2 Report
The current version manuscript is substantially improved. the authors polished the language and included additional analysis like functional annotation.
some few additional amendments are suggested:
Botanical names must be italicized
Improve on the quality of Figure 2
The figure on distribution of TcGRAS genes on chromosomes has no title...why did the authors combine the titles in one figure?
Line 422-431 is not relevant in the introduction. Authors should move these to introduction.
line 446-448, can authors provide evidence ? Additionally, authors have not convincingly accounted for the differences in TcGRAS in the different sub-groups on the phylogenetic tree
check references and adopt one referencing format. Some journal names are italicized, some not.
Please look out for additional comments in the attached manuscript
